# “We Cause a Ruckus”: Exploring How Indigenous Youth Navigate the Challenges of Community Engagement and Leadership

**DOI:** 10.3390/ijerph19159542

**Published:** 2022-08-03

**Authors:** Robert P. Shepherd, Treena R. Orchard

**Affiliations:** 1School of Public Policy & Administration, Carleton University, Ottawa, ON K1S 5B6, Canada; 2School of Health Studies, Western University, London, ON N6A 3K7, Canada; torchar2@uwo.ca

**Keywords:** community engagement, Indigenous youth, leadership, mental health, resilience, strengths-based approaches

## Abstract

Using qualitative data from an interdisciplinary research project about mental health and community engagement with Indigenous youth in Kasabonika Lake First Nation (Ontario, Canada), this paper explores the factors that constrain and facilitate their ability to contribute to the well-being of their community. Case studies are employed to demonstrate how the youth navigate complex social and structural conditions within the context of on-going colonization through federal and provincial governance arrangements, to make a difference in the place they call home and forge unique in-roads that reflect their generational realities and aspirations. The paper contributes to ongoing discussions related to mental health, self-determination, and resilience research.

## 1. Introduction


*A health worker said that resilience and strength come only when a crisis occurs…It’s hard to get people out for workshops, activities, and community engagement. The top mental health problem among youth is anxiety, and this group doesn’t want to talk about what’s bothering them. Youth tend to be shy to outsiders because there is no trust. There needs to be more involvement in the community to build that acceptance ([1] January 2019]).*


Indigenous youth under the age of 25 are among the fastest growing populations in Canada, comprising approximately 50 percent of total on-reserve populations [2]. Many of these youth experience complex forms of intergenerational trauma, structural violence, and health as well as social adversities stemming from colonization [3,4], including housing insecurity [5], food insecurity [6], and increased risk of substance use [7], sexual violence [8], and mental health challenges [9,10]. Alongside research about the socio-structural and political conditions that prevent Indigenous youth from participating equitably in society, there are studies that explore the remarkable knowledge, creativity, and strength-based solutions [11,12] youth possess as they express who they are, contribute to the well-being of their communities, and define their own pathways toward resilience [13,14,15,16].

Resilience in this setting refers to the adaptive approaches used by youth to respond to crisis situations and everyday life shaped by structural and on-going traumas associated and connected with colonialism that include engaging with cultural activities and teachings, positive peer and family relationships, and having a robust sense of self as well as wellness [12]. There are many socio-cultural and political factors that both facilitate and complicate youth-driven resilience initiatives. Among them is the role played by local power relations, political leadership, and targeted economic opportunities in remote communities that are rarely assigned for the enhancement of youth-led programming. This exclusion stems from many complex factors: administrative dependence on the state that is bred into and often fractures the flow of everyday life in Indigenous communities [17,18,19]; and the tensions many adults feel about wanting to support their youth but having few viable opportunities to do so [12,20]. Cultural dissonance between generations and a tendency for some older community members to discredit the abilities of younger people amplifies the challenges facing Indigenous youth [21,22]. These strained intra-community dynamics can reduce their capacity for resilience and impair the confidence they have in their ability to contribute creatively, socially, and politically to the health of their communities. Indigenous youth want to participate in decisions about their future in ways that draw upon their unique cultural resources and reflect their generational experiences [23,24,25].

Engaging Indigenous youth in research in meaningful ways requires a collaborative, strength-based approach whereby researchers are not leading but facilitating or supporting youth themselves in their efforts [26,27]. Flexibility, ongoing support, and engagement styles that correspond with how youth communicate over time are identified as important to relationship building and creating conditions of trust and engagement in Indigenous contexts [28,29,30,31]. Indigenous youth are not simply the beneficiaries of knowledge loss, they actively produce their own cultural knowledge [27,32] that can be used to enhance the “alimentary infrastructure” of their communities [33]. Firmly rooted in sustainable, modern practices of governance and the power of storytelling, this kind of infrastructure is life-giving and positions youth as generators of restorative change in their communities and within themselves.

This paper contributes to these dialogues about honouring and supporting youth-driven community initiatives. It shares findings from Kasabonika Lake First Nation, where we have collaboratively undertaken research on several youth-led projects designed to improve mental health, enhance community engagement, and create governance infrastructure that reflect youth perspectives. Although opportunities for young people to meaningfully participate in the social organization of their community are limited, they leveraged various forms of social capital, such as informal relationships with role models and mentors, to engage with local leaders and advance a series of initiatives with the support of external resources. Illustrative case studies are used to explore how the youth employed these activities as a way to generate trust among themselves, other research team members, and their community.

The paper is divided into three major sections, beginning with a detailed account of methodological underpinnings and approaches undertaken during the project. The second section provides the study findings in the form of four case studies that highlight youth-led social activities and infrastructure-related projects. The final section provides points of discussion that inform future research about mental health, youth leadership, and resilience.

## 2. Methodology

### 2.1. Study Setting

Kasabonika Lake First Nation is a fly-in community in northwest Ontario, Canada located along the Asheweig River, which is a tributary of the large Winisk River that flows north into Hudson Bay. The community was originally affiliated with Big Trout Lake First Nation, but in 1979 received its own reserve status. Signatories of historical Treaty 9, Kasabonika Lake is part of the Shibogama First Nation Tribal Council and local health services are administered through the Thunder Bay District Health Unit. Band office operations include a Northern Store (operated by the Northwest Company, Winnipeg, MB, Canada), Canada Post station, hotel, restaurant, and scheduled as well as charter flights are serviced through the Kasabonika Lake Airport. There is also a community centre that is used for hockey, skating, bingo, dances, feasts, and other social gatherings, along with a church that offers traditional, Anglican, and Gospel worship services.

Home to approximately 700 people on-reserve, Kasabonika Lake First Nation is a young community, and many members speak and maintain strong ties to the Oji-Cree language. As with most reserves across Canada, community members struggle to access the basic social determinants of health with respect to safe housing, quality education and social services, and basic economic infrastructure. They also have inadequate healing and crisis services to help combat the effects of intergenerational trauma stemming from residential schools, the 60 s scoop, and the structural violence that affects everyday life.

### 2.2. Epistemological Approach

Research is a politically charged set of activities, usually understood through Western ways of knowing that tend to be extractive in nature [34]. The research team, however, understands research as more inclusive, which is fundamentally about relationships, mobilizing Indigenous voices, and honouring Indigenous knowledge [16,34,35,36] as well as that which flows from Western ways of knowing rooted in observational techniques [37,38]. Our project was informed by a phenomenological orientation, which is a relational and social constructivist perspective that acknowledges one’s own research assumptions and biases. Under this orientation, there is no singular reality or interpretation of the world, but there may be essences that are shared among multiple individuals [39]. Phenomenology as a research approach allows for the exploration of how individuals define their reality and experience the world, including social norms, religion, social ethics, interpersonal relationships, and subjectivity [40]. It also aligns with our reflexive positionality and research praxis, which involved listening closely to how the youth and other stakeholders articulate their lived experiences, recognizing that “there are multiple sides and multiple expressions of possibilities active in any situation” ([41] p. 116).

This phenomenological approach is activated through Two-Eyed Seeing, developed by Mi’kmaw Elder Albert Marshall and defined as: “To see from one eye with the strengths of Indigenous ways of knowing, and to see from the other eye with the strengths of Western ways of knowing, and to use both of these eyes together” ([42] p. 335) and [43]. Acknowledging the power differentials between team members [44] was a practice the researchers adopted in fieldnotes, phone calls, and private moments of reflection. As Marker [25,45] argues: “To really learn about Indigenous communities is to learn about oneself, and researchers are not trained for this encounter.” This was a vital way of learning how various community and research-related constraints affect the youth, as this fieldnote [1] excerpt from one in-person visit to Kasabonika in the Spring of 2020 illuminates:


*We are seeing more and more each visit, which in many respects is an indicator that many in this community are letting us in to see what we see. There is still very much a distance that I doubt will ever be closed, but many are willing to trust us up to a point. The youth reach out, again to a point, but we are able to leave Kas and the youth cannot. I can’t help but think there is a certain resentment about that—not that we are privileged in some way (which I’m sure is part of it), but because the work stalls when we are not there. They need support, and while they get some for basic things, the more difficult work of mediating adult relationships stops.*


### 2.3. Research Approach and Data Collection

The project upon which this paper is based is a Social Sciences and Humanities Research Council of Canada (SSHRC) funded, Indigenous Youth Futures Partnership (IYFP) [46] project that took place between 2016 and 2022. The aim of the project is to foster Indigenous youth resilience and help empower youth to prosper as leaders in their communities [47]. Our abilities to understand the realities of life for youth in a fly-in community have been challenging. The IYFP trips we did together were squeezed into busy research and community schedules alongside the obfuscations brought on by the COVID-19 pandemic, which brought all in-person visits to a halt. Building strengths-based relationships takes much time, energy, patience, and trust-building, which has been complicated by not seeing one another face-to-face for two years. In the interim, digitally mediated communications have served as a lifeline to maintain our relationships, and also support observations into what is happening in our respective life worlds.

Engagement with the youth occurred over the course of 30 visits of 3 to 4 days each between June 2016 and March 2020. During the pandemic, March 2020 to March 2022, contact was maintained on Zoom and Facebook Messenger. During this period, the team organized roughly six virtual visits annually. Four youth (3 female, 1 male) self-selected as being interested in working as Youth Apprentices (YAs) on the project, and a fifth male youth also took part intermittently starting in the summer of 2019. Apprentices led discussions with the research team and youth on the activities they wished to organize and provided the essential leadership to ensure these activities were carried out. Several community mentors worked informally and formally with youth programming in the community. They helped the youth and research team liaise with leadership and provided considerable advice, help, and resources. These individuals included the Director of the Jordan’s Principle Program, Director of Brighter Futures Program, Director of the Choose Life Program and staff, Policing Services, Director of the Resource Centre Program, Director of Education, and Director of Health. Staff from the Sioux Lookout First Nations Health Authority (SLFNHA) and Shibogoma Tribal Council also participated by organizing sporting, instructional or fieldtrip activities that encouraged youth to create friendships and assist the team to create opportunities for youth engagement.

Each visit to the community commenced with a formal request to the Deputy Chief responsible for youth programs. This individual was also the main conduit to community workers, leaders, and service providers. The agendas of our visits were often informal as these were dependent on the YAs. The aim was to work directly with youth and the community in ways that they identified as being useful within the context of the team project and emergent community issues-a form of embedding [44,48]. The researchers engaged in multiple interactions with an array of people who come into the community and residents (i.e., construction workers, teachers, traditional healer, hotel personnel including clerk and cleaning staff, shuttle workers, airport workers), which generated rich insights from diverse social locations, affiliations, and experiences. Formal meetings with Chief and Council, health workers, education administrators, and mental health treatment workers were also undertaken to maintain support for the project, and to elicit their commitment to youth projects that formed the basis of our case studies.

Fieldnotes were prepared by all team members and collated at the end of each field trip amounting to approximately 20,000 words combined. Compared to lab-based research activities, longitudinal surveys, and audio-recorded interview materials, typically regarded as the gold standard of data recording techniques, qualitative field notes are often viewed as ‘uncooked’ knowledge of questionable validity and reliability [49,50,51]. These positivist claims are challenged by social scientists, particularly ethnographers, who have always relied upon these vital repositories of information in field settings where the recording of information and delivering lengthy surveys are not suitable ethically or culturally [50,52,53]. Trying to arrange individual interviews, for instance, in a socio-centric cultural environment where indirect forms of communication and observation are central to knowledge acquisition [54] would be culturally insensitive and would lack beneficence [34].

Neuman [55] identified six types of fieldnotes that are commonly used in qualitative data collection: jotted notes (short memory triggers), direct observation (written immediately after leaving the field), inference (reflecting social relationships, emotions and meanings), analysis, (methodological strategies and theoretical notes), interview notes (information about interview location and interviewee), and personal journal (personal feelings and emotional reactions). Each of these forms of notetaking were used at different times throughout each of the visits (i.e., planning for a visit, throughout a visit, and post-visit reflections), and the content of our notes shifted depending upon the events we witnessed. Single-person and multi-person visits were captured using fieldnotes that aligned with the aims of direct observation, inference, analytical, and personal fieldnotes. Each team member’s notes were consolidated into a master document within a day or so of the visit, which represented a rich, thoughtful, detailed record of the timeline of events and the unique dynamics of the different field visits. This task was shared by team members, who took turns doing the consolidation. These techniques are consistent with narrative analysis [56], which attempts to bring together different policy or observational “stories” into a relevant meta-narrative.

Alongside the various situations and conversations that we recorded in notes, understanding the lived realities and perspectives of youth in Kasabonika Lake was paramount. We focused intently on the observations that demonstrated and enhanced our understandings of youth resilience and how they navigated and resisted local power structures while creating their own path. Our observations highlighted and emerged from the intersection of various forms of socio-political engagement, local leadership structures (i.e., generational uniqueness, kin networks), youth creativity (i.e., use of limited resources and art forms), and power relations (i.e., youth exercise limited power and navigate relationships with senior leaders). The youth’s success in these endeavours demonstrated their agential capacity and their ability to generate greater trust and confidence from senior leaderships as manifested, for example, in the increasing importance assigned to youth voice by community leaders over time.

### 2.4. Data Analysis & Ethical Issues

Data gathering was organized according to key principles for thematic analysis developed by Braun and Clarke [57], beginning with multiple and close readings of the fieldnote data that helped elucidate youth perspectives and experiences: community engagement, leadership, resilience, and their own identities as youth who face considerable socio-cultural, political, gendered, and inter-generational struggles. These were further specified by drawing upon the following indicators developed by Snow et al., [58]: Indigenous identity development; Indigenous paradigmatic lens; reflexivity and power sharing; critical immersion; and participation and accountability. The data were then organized into master code files according to these indicators and other emergent findings, which were then reviewed using line-by-line coding (and validated with NVivo software) to further refine the data related to the primary study aims. 

Data analysis was guided by critically oriented feminist and post-colonial approaches that highlight the ways that structural factors and the conditions of everyday life can both constrain and offer productive avenues of resistance among the youth as they acquire generational knowledge about community dynamics, leadership approaches, and their own way of being in the world [4,11,59]. These approaches align with our epistemological orientation, especially navigating the interface between Indigenous and Western research to generate findings that reflect the interests, values, and priorities of Indigenous peoples [34,60]. We tried not to be overly predictive of findings from the cases in the Western sense so as to allow Indigenous voice to come through as it emerged.

The case studies selected were determined by team members, along with the supportive insights of the youth, as being especially evocative examples of leadership, engagement at the local level, and the challenges youth encountered as they sought to stake a claim in their own lives and the organization of their community. 

In terms of theoretical positionality, the authors come from distinctive yet overlapping and complementary fields of expertise and orientation. The first author is a public administration and program evaluation specialist who uses contemporary mixed methods approaches such as theory-based analysis in a post-colonial context related to matters of governmental policy formulation, and with structuring work with Indigenous and vulnerable populations. The second author is a medical anthropologist and feminist scholar who employs post-colonial and intersectional theory related to sexuality, gender, and health in her work with vulnerable communities and within the context of digital culture.

The project received ethical approval from the principal investigator’s home institution. The project also received approval by the Sioux Lookout First Nations Health Authority (SLFNHA) as the main partner on the SSHRC partnership grant in May 2016, and by Kasabonika Lake First Nation in September 2016. Youth apprentices in Kasabonika Lake are compensated for their time by the project, and in-kind contributions by the community are recorded under the grant.

Our sample of youth participants (*n* = 4–5) was small, but this did not deter our project aims. There are very few youths in the community, who have graduated high school and had the skill set to contribute to the project as youth apprentices.

## 3. Representative Youth Projects

Our primary data for this paper are presented using an illustrative case study approach [61] that relies on both propositional and tacit knowledge in keeping with our epistemological approach. We discuss four cases or activities we undertook in partnership with the youth which demonstrate how we came to know the youth and understand their perspectives on what they want for their community. These activities were identified by the youth themselves as important priorities. However, there were several other activities that also held importance. We identified these cases or activities on the basis that the youth thought these raised notable contextual challenges, creative solutions to problems, and observable wins they felt should be highlighted over other activities that were ongoing and required a longer-term view.

The cases also illuminate how we came together as a team to exchange ideas, convey knowledge, and create conditions of trust [29]. Two of the case studies focus on community-level initiatives the IYFP team developed to enhance community engagement and encourage social change. The other two cases highlight initiatives designed to support structural-level change. The community-level initiatives fostered a sense of cohesion among the youth and the researchers as well as between the youth and other community members particularly youth. The structural-level projects reveal how the youth sought to better understand and mobilize formal and informal power structures in the community, including decision-making bodies and exchanges with elders and other informal leadership agents.

When faced with complex socio-cultural and economic conditions that impaired their abilities to meaningfully contribute to local community dynamics, the youth respond in a myriad of ways. Sometimes they pulled pack, while on other occasions they cultivated their own support networks to help one another, other youth, and give back to their community. At other times they aligned themselves with powerful local informal leaders to help secure the support needed to achieve their aims. Speaking specifically about how they garnered the support of other youth, one YA said: “We’re the group of youth who are most known in the community...we’re close to people who are well known in the community…we cause a ruckus, that’s why we’re noticed”. The strategic value of being “favourites” of the local Chief & Council and having family members who participate in important decision-making processes within the community were also cited as being of direct benefit in helping achieve their goals. As demonstrated in the case studies below, when faced with precarious social conditions and age-related forms of exclusion, the youth flexed their resilience muscles and were successful in scaffolding the cultural capital at their disposal, including that which is relational and mediated through kin networks, to achieve their own ends.

### 3.1. Case Studies 1 & 2: Social Change Projects

These case studies feature initiatives that the youth developed on their own and in partnership with team members, including community game nights and making winter snow angels, along with organizing a Valentine’s Day dance, the first event of its kind in Kasabonika Lake. These activities enhanced social cohesion among the IYFP team and helped position the youth as recognized leaders within their community. They also illuminate how the youth apprentices successfully engaged with different stakeholders to generate social change, especially for other youth who have few opportunities to participate in fun activities that are designed just for them by other young people.

#### 3.1.1. Game Nights & Snow Angels

A key aim of the IYFP project was to create culturally relevant, fun activities for community youth in order to keep them busy and to help cultivate peer-based social cohesion. During a team zoom meeting in March of 2020 about the YAs accomplishments to date, the following question was posed: ‘What was the turning point in the project?’ Referring to when the youth felt like their work had gained momentum and was garnering increasing participation. They all mentioned game nights and snow angels. Since these events took place in succession, between December 2019–January 2020, we have grouped them together here. Prior to launching these two particular activities, the YAs brainstormed about potential activities that could help get youth and others engaged in the project. The enormous list, illustrated below in Table 1, reflected their creativity and knowledge of their cohort as well as their community. Potential activities included:

Using Facebook, which is accessed extensively in the community, the youth began sending out messages about activities just for young people. For the first event, they held a board game night, which was low cost, easy to organize, and something youth of all ages enjoy. It was held during the evening, which is when young people preferred to hang out together, and light snacks were provided. The YAs documented the event and uploaded pictures to their IYFP Facebook page as well as their individual accounts. The snow angel activity was spear-headed by one of the YAs, who posted instructions for the game on Facebook and those who took part would go to the designated spot, make fun snow angels, and share photos.

From the informal participant feedback, both of these events were enjoyed by the community youth (*n* = 15 for each), and they were each held on several occasions. The activities doubled as opportunities for youth to cultivate safe spaces to talk about the challenges of life in Kasabonika and coping strategies for dealing with different aspects of mental health (i.e., other than substance use and self-harm). A core of youth took part in each activity, but the composition of the attendees ebbed and flowed over time to reach a diverse set of community youth. Although positive, the YAs discussed the tensions between wanting to support their peers who reach out to them, needing it themselves, and the stigma associated with obtaining mental health care:


*The kids are of all ages and [one YA] who spoke with two of them last night, said they felt like there was no one to talk to. That hit the YAs hard, and they were like ‘we understand, but we’re here too’. There was a pause in the conversation, and I don’t think any of them have seen a counselor or if they have, maybe for only a couple of crisis-related sessions. When I asked if they’d seen a psychologist or someone like that, [the YA] said: ‘NO, they’ll say I’m crazy’ ([1] March 2020).*


#### 3.1.2. Valentine’s Day Dance

One of the most significant events facilitated through the IYFP was the Valentine’s Day dance held on 14 February 2020. Kasabonika had never seen a youth-driven initiative of this scale and it involved months of careful planning and fundraising. Selected as a way for people to celebrate one another on a day devoted to love, the dance was an incredible feat of resilience and community engagement. It also involved the youth apprentices stepping into leadership roles that were new for them as they mobilized resources, spread the word, and found ways to create a welcoming environment for the participants. One of the challenges identified early on was that many people in Kasabonika did not own formal dress wear, which could prevent them from attending. The YAs responded by soliciting donations for formal wear online and with other IYFP team members during a meeting in Ottawa attended by the authors. An array of dresses, shoes, and men’s wear in different sizes were purchased at a second-hand boutique and displayed on Facebook for community members to claim prior to the event. Additional clothing and accessories donations helped tremendously in creating a community clothes closet for the dance. Floral decorations were also purchased to help cultivate an ‘enchanted forest’ theme, which brought considerable colour and flair to the event.

Another strategy to generate buy-in was to offer prizes for the attendees, who were organized into four main categories: Kings and Queens, which included women and men over the ages of 16, along with Princes and Princesses, which included boys and girls aged 15 and under. These categories each had corresponding prizes of significant monetary and recreational value: King & Queen- 1st—TV and $100; 2nd—$200; 3rd—$150; and Prince & Princess: 1st—Nintendo Switch and $75; 2nd—$150; 3rd—$100. The funds to cover the cost of the prizes were generated by a small grant that the Yas assembled for a regional Indigenous service provider, who were named as sponsors of the event and thanked formally by the organizers. The dance took place on the same night that a bingo event was being hosted through the local radio station, which is normally a very well-attended activity. However, many people chose to attend the dance instead and in total approximately 75 people participated, including youth mentors and program leaders.

The YAs were thrilled with the event, which they indicated had helped increase people’s confidence and encouraged some community members to wear more fashionable clothes. Reading love letters was the main contest participants entered to win prizes, and this feedback from one of the attendees sums up how touching the event as well as the community pride in the YAs for their hard work: “I almost cried when they read their letters and sang their song lol. So nice listening to them, you get this proud feeling for them for stepping out 
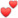
 Good job” ([1] February 2020).

### 3.2. Case Studies 3 & 4: Structural Engagement Projects

The following two projects provide a glimpse into how the youth have attempted to influence formal power dynamics within the community in ways that demonstrate their creativity, respect for others, and their humility in gently bringing their community along with them. These initiatives required some support from team members, in order to tap into experience working with formal power structures.

#### 3.2.1. Youth Space Development Project

An important objective of the IYFP project is to engage with the youth on issues and projects that could influence the way youth relate to formal power structures in the community. One ongoing project is the creation of a “space of their own” where youth can attend to activities on their terms. As one young boy said: “Having our own place would go a long way to giving us a place to go, hang out, and be ourselves” ([1] February 2018). These discussions were first raised with program directors and youth when the team visited the community in February 2018, but they have yet to lead to concrete outcomes. In the absence of a youth centre, the youth said they call the canteen in the arena their space, but this is largely under the control of Chief and Council with time in that space tightly controlled. The research team decided then that questions related to a youth space should be ongoing.

In our efforts to keep this issue moving forward, when we visited Kas in April 2019 we asked the high school to carry out a survey about the needs of youth and how these could be addressed. The responses are captured below in Table 2. These responses were validated by the youth apprentices in December 2019 in discussions about what they would like to see in a youth designated space. Of particular note between the high school students and youth apprentices was the need for youth programs.

Between May and November of 2019, plans were underway by Chief and Council to create a multi-purpose centre, including a youth space, in partial response to the requests of youth. However, as often occurs, the “adults” create the plans, but do not always include the youth in them. The youth expressed the desire to have a separate space that they could manage on their own, away from an “adult-managed” facility. However, the community’s plans for such a complex appeared to fizzle out by the end of the year due to provincial budget cuts, but youth concerns remained [62].

Undeterred, the youth began advocating for their own space in May 2019 and sought the support of the deputy chief to either use space in a senior’s complex or configure six donated Hydro Ontario worker pods into a building on an abandoned construction site. The pods could be connected, and the youth apprentices took great pride in creating a building plan for them. The youth in the community were finally seeing some progress not only on the site but on plans for youth programming by spring 2020 ([1] June 2020). Over much of the remainder of 2020 and early 2021, many of these plans underwent significant change, but youth continued to be assured that their own space would be created on the abandoned construction site once it was no longer needed. As of March 2022, a commitment was made by Chief and Council to use the construction site for a youth centre pending the completion of a new school and the construction site is needed to store materials and equipment. Although youth were asked to wait for the school to be built, the project highlights youth’s growing resilience to roll with the shifting politics in the community but with the added confidence to hold their elected officials to their promises, despite an inconsistent track record.

#### 3.2.2. Youth Participation in Community Decision-Making

Another important project spearheaded by the youth apprentices and encouraged by the IYFP team was to find a way for the youth of Kasabonika Lake, aged 12–26, to regularize formal engagement with Chief and Council on various youth matters, including representation on Council. In March 2020, the youth apprentices and the team met to revisit various projects, including the youth centre and potentially building a library, and holding other events that built on the success of the Valentine’s Day Dance. The research team held informal sessions with the Chief, Deputy Chief and various program directors to provide updates on the IYFP work with the youth and conveyed the argument for council representation. Playing this role can be awkward, but the IYFP role in the community was increasingly seen as an advocate which made the interactions more straightforward over the three years spent cultivating relationships. In response and to our surprise, the Chief committed informally to having two youth representatives on Council and promised to hold a special meeting on the matter while our team was in the community.

The Chief called a council meeting dedicated exclusively to youth matters, which held their promise as there appeared to be a commitment for youth representation by the executive committee of Chief and Council and the senior program directors. Two youth apprentices joined the research team at the meeting where presentations were made by various directors on how much was being done “for” youth, including the purchase of canoes, hunting equipment, and the like, but it was clear that these announcements were intended to showcase their accountability more than respond to or include the needs youth could communicate themselves. Arguments were made for and against representation, mostly by the “bureaucrats,” who argued that community consensus was needed on amending the configuration of Council—an *Indian Act* requirement that does not apply to special representatives such as youth ([1] March 2020). There were also arguments of distrust, again by unelected officials, in youth abilities to manage such responsibilities, which was disappointing on so many levels to the extent that the youth in attendance disengaged again. In the absence of direct representation, the team and youth suggested that a youth council be organized instead, which was accepted by Council, but would have to be discussed at a future date—then, the pandemic lockdowns hit a few days after our return trip. Upon our departure, we reflected:


*The AP’s need support, and while they get some for basic things such as the Valentine’s dance, the more difficult work of mediating adult relationships stops ([1] March 2020).*


In June of 2020, the researchers and youth apprentices crafted a letter to Chief and Council requesting the establishment of a youth council comprising ten youth who attached their names to the letter, promising “to have weekly meetings to maintain our program and to ensure there are updates on major priorities, including the construction of the Youth Centre… and hosting youth events in the community, specifically projects for youth by youth”. The youth council was formally recognized by a band council resolution in August 2020. The council has met over Zoom on and off since then, again showing their commitment to bettering their community and moving forward with projects, albeit more slowly during the pandemic. Most meetings, approximately four per year (rather than the promised weekly meetings) attract between 4 and 8 of the original ten signatories to the band council resolution.

## 4. Conclusions

Speaking at an in-person IYFP conference in January 2020 at Carleton University in Ottawa, Canada, one of the youth apprentices explained that “we cause a ruckus” to highlight how youth are engaging with their community. The statement was profound in a few ways: it captured the reality that youth in the community have to be persistently loud and unrelenting in order to be heard about the community they will soon be inheriting. It also demonstrates that many youths are weary of being powerless against a constant wave of decisions that continue to work against their engagement and participation in local decision-making structures. Despite the rhetoric of local leaders, Indigenous youth in Kasabonika Lake are rarely actively or seriously consulted for their input on major or even minor issues of governance, resource allocation, or ways to improve the lot of youth, which is consistent with the experience of many First Nations communities [59]. Although there have been nods to youth participation through the creation of a youth committee, it takes significant commitment both in the form of leadership energies and community supports such as funding to make it part of local governance.

Despite the challenges, there are several observations that could be made about the poor state of local First Nations government and governing under the backdrop of the *Indian Act* [63]. It is increasingly apparent that many Indigenous youths in this community and others participating in the Indigenous Youth Futures Partnership are becoming more confident about what they can do. They want to be heard, and their message is that the past need not define the future, that change is possible, and that this change has to come from the next generation which believes it does not have to be encumbered by past external authorities and decision structures. They are resisting their exclusion from decision-making by taking it upon themselves to create the conditions for youth engagement on their own terms [27]. This attitudinal change embodies the necessary first steps toward creating the conditions for resilience against indifference or neglect. Deciding that change is necessary and that youth have a role to play in that change is fundamental to building identity and the space needed to play their part [10,64]. This paper suggests that for resilience to become something more than conceptual, it has to come from cognitive and attitudinal acceptance and a desire to create change.

As part of this attitudinal shift, Kasabonika Lake’s youth have demonstrated a desire not to displace current power structures, but to find new ways of making them more open to youth participation and ways of doing things, and in a manner consistent with their aspirations. Youth have demonstrated that they want to contribute to their community’s future in a way that is respectful of their elders, while nudging their leaders to consider different ideas.

Incremental change from exclusion to inclusion takes time and local efforts are now needed at addressing larger structural issues such as those related to the traditional *Indian Act* governance model that limits power to Chiefs and Council and their bureaucracies, which often operate at a distance from the people they purport to serve [65,66]. There is also intergenerational trauma caused by residential schools and other visible and tangible colonial projects of the past that divorce community members from their traditional ways of doing things everyday, including that current leaders see structural barriers to change and inclusion where youth increasingly do not. Breaking down these structures and associated attitudes associated with the *Indian Act* and of colonial approaches to doing things are not merely institutional in the sense that one-size-fits-all approaches impose bureaucratic, political and social requirements as conditions for financial and other support [67]. They are also cognitive in that community leaders may fear or are reluctant to embrace other approaches or models of inclusion that reduce their power or influence that comes from band government elections and federal and provincial funding. More importantly, these structures provide cover for limiting the participation of youth in community decisions [63,65]. In Kasabonika Lake, many youths have communicated in various ways that these structures no longer have a hold on their futures, and they are wanting change from within their community—the impetus for change can no longer come from outside. In effect, youth regard structural change as internal to their community. The effects of residential schools and other traumas are in the past for youth, but they understand that intergenerational trauma caused by these intrusions has to be repaired in the ways of their community.

The four cases demonstrate the singular message that the youth of Kasabonika Lake, exemplified by its leading youth apprentices’ voice, are able to come together and build the alliances they need to organize and execute their projects aimed at addressing trauma and creating the conditions for resilience against such trauma in the future. Creating a “ruckus” has meant taking the risks needed to insert themselves in political processes in order to generate momentum for the work they believe is needed to engage youth and invest in their future. They may be excluded in many respects from the formal sources of power by virtue of their age, but the youth have demonstrated an ability to create kinship ties and political alliances in ways that do not always require formal authority. As indicated, youth are increasingly ignoring externally imposed band government systems, but instead recognize traditional ways of governing that include the voices of women and traditional power holders such as knowledge keepers. In other words, they are learning to mobilize various sources of social capital in informal ways and carve out their own leadership in relation to various formal power structures such as the band office and its various formal concomitant programs and services.

Our team adopted a strengths-based approach that documented not only the hardships faced by the youth but also their creative solutions to the challenges encountered [64,68]. The results observed have been encouraging that with sustained support of external resources such as the research team and the commitment of informal leaders in the community, trust can be established with youth in ways that create momentum for youth initiatives. Youth can also be encouraged to take a leadership role in their own projects drawing on the experiences of age, generational uniqueness, and kinship networks that offset their powerlessness. As demonstrated by the projects, youth have the ability to innovate within their community, and draw on their networks and supporters to provide resources that do not rely on formal programs and services. They are able to create their own influence through generational issues such as the increasing role of technology or the importance of various social issues including environmental protection or sexual identity through art, dance or social awareness events. Most importantly, they are creating their own power that is alimentary and not simply oppressive. When youth do assert themselves through such projects, it demonstrates to other community members that there is an appetite for change, that there is hope for the youth, and that there is value in investing in these projects and activities. However, as the youth apprentices have learned, getting these events moving takes effort and there is strong resistance that constrains new ways of creating the conditions for change.

Our findings align with current research about resilience among Indigenous youth, which demonstrates the importance of engaging with cultural activities and teachings, positive peer and family relationships, and having a robust sense of self as well as wellness [12,23]. The unique role played by age as a determinant of resilience should be highlighted, which demonstrated that despite being excluded within official power structures because of their socially invalidated status as youth, the youth cultivated their own powerful sense of sub-cultural community and fostered meaningful cultural activities that provided an outlet for socialization that was youth-specific and benefitted the whole community. This accomplishment demonstrates how they are thriving despite internal barriers as well as the broader forces of marginalization stemming from the ongoing colonial process.

## 5. Next Steps

These cases in Kasabonika Lake demonstrate that with encouragement, consistent and supportive formal and informal leadership, youth are able to see what is possible and make change happen within the context of their own lives. These rewards are cumulative and extend inter-generationally, with younger youth picking up where others have left off [67]. The future of Indigenous community health is dependent upon investing in youth, fostering meaningful leadership opportunities for them, and trust-building. The youth on this team have consistently demonstrated that they possess the agential power needed to enhance their community and carve out important roles for themselves within it. These findings align with other research on youth engagement [5,12,20,26] that indicate how engagement must originate from youth and be on their terms. As researchers, our role is to understand the issues they believe are important and take cues from them about how to best support project activities through their cultural and generational lens, which is the essence of Indigenous Youth Futures.

Supporting additional research in remote Indigenous communities that is attentive to similar issues can make a difference for sustainable socio-cultural change. The role of such macro community factors such as kinship ties, religion, tradition, protocols, economics and social determinants of health are profound but each of these hold differing degrees of significance on matters important to youth. Although there are many statistical studies that attempt to understand causation between such factors, qualitative narrative adds depth of understanding to what matters to youth and why. Finding ways to access these narratives that are so dependent on trust-building is an important focus for future research in this area.

## Figures and Tables

**Table 1 ijerph-19-09542-t001:** Brainstorming Activities for IYFP Programming in Kasabonika Lake.

Community contest to determine the name of our project Beach party with volleyball Game night (board/video games) Movie night Music classes/art classes Digital media and drone flying Fishing, canoeing, kayaking Bonfires, wiener roasting Build a raft and see if it works; recycle rafts Blueberry picking and bonfires Trails/mountain biking Spotlight/Hide-n-Clap game Old settlement day trip Sports tournaments	Documentaries about Kas LGBTQ2S+ sensitivity and safety training Bingo Make your own go karts Rabbit snaring Partridge hunting Exercise course/trail Obstacle course Slime-making Shoreline fishing Pig roast Career Fair with employers in Kas Colouring contests Drawing/painting classes Painting houses/buildings Clearing space for benches and fire pits for youth gatherings

Source: [1]: December 2019, January 2020.

**Table 2 ijerph-19-09542-t002:** Youth-Identified Needs in Kasabonika Lake By Age.

High School Responses (*n* = 40)	Youth Apprentice Responses (*n* = 4)
Kitchen/Canteen Video and board games Equipment for traditional activities, such as trapping, hunting, fish nets, arts, sewing Big screen TV/Theatre Musical equipment Pool table Gathering space for youth discussions and events, such as training Computer room Special programs to address the “troubles” of youth such as “kids who have suicidal thoughts,” substance abuse challenges, and trauma Outdoor gathering space	Club rooms for arts programs, etc. Outdoor gathering space with fire pit Parking and picnic areas Games room Kitchen/canteen for youth gatherings Library capacity Washroom facilities and laundry Specialized spaces for youth counselling Organized programs for substance abuse, intergenerational trauma, depression, alcoholism Place to store equipment (canoes, nets, etc.)

Source: [1]: Fieldnotes, April 2019, December 2019.

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
