# Peer review of "“We Cause a Ruckus”: Exploring How Indigenous Youth Navigate the Challenges of Community Engagement and Leadership"

_ijerph, 2022, doi:10.3390/ijerph19159542_

Round 1
Author Response
We wish to thank reviewer 1 for their comments. We found many of them very helpful to strengthen the paper. See our detailed comments.

Reviewer 2 Report
The authors of this manuscript present a multiple case study detailing the factors that constrain and facilitate Indigenous youth engagement, participation and empowerment in Kasabonika Lake. This paper has many notable strengths. Addressing this topic area has critical implications in creating voice for youth, exemplifying leadership amidst internal and external power structures, and the development of critical alliances needed to overcome challenges. Overall, the manuscript reads well and is a great contribution to the literature. With these strengths in mind, the following points are offered to assist in the overall quality of this written product.
- Overwhelmingly this manuscript has many run-on sentences that often span more than 3 lines of full text. See lines 44-48, 51-56, 57-68 (representing only 2 sentences) as examples, but there many more throughout and far too many to list in detail. The content is very important but the sentences would read much better if they were shortened.
- Line 106 - I suggest that the author phrase the definition of “research” that is used here. Traditionally, “research” is grounded in Western ways of knowing but the author is attempting to expand or reframe this definition to be more inclusive. Rightfully so, this suggested revision needs to be made more clear. e.g. The author should state the Western perspective (often extractive means of engagement) and then rephrase with the more inclusive, relationship building, honoring other ways of knowing/Indigenous knowledge perspective.
- Section 2.4 – I suggest that the author add substantial detail to this data analysis section. Please detail the type of case study that is being presented; is this considered to be historical, problem-oriented, multiple, intrinsic, or illustrative? The author mentions the use of interpretive critically oriented frameworks but does not provide details for what these mean or how these are used. Additional information should be included that details the data analysis team and how analysts were able to find consensus while contextualizing the findings? Were all 6 different type of field note types utilized in this research, and if so how were these organized and synthesized? How were the findings organized and how were themes prioritized across the different field note types and data collection methods? Did you use any type of qualitative data management software to help organize the data? How and why were these specific case studies prioritized? Perhaps the authors could include a matrix of some sort – multiple case studies often have matrices to help the reader understand the analytic process, how themes were generated, and how summaries were developed.
- Line 223 details youth participants (n=4-5), why is there a range and not an exact number of participants?
- For each of the case studies, please detail the reach of the activities and provide an estimate of how many youth attended or participated in the efforts detailed. This can be useful in further emphasizing the great impact of these youth efforts.
- Line 269 – It is my assumption that “YA” represents “youth apprentice”? Please define this term.
Author Response
We wish to thank reviewer 2 for their comments. Many of these were very helpful. However, we wish to highlight in our report the limitations of depending too heavily on Western methodological approaches. We tried very hard not to make the methods sections overwhelming as to contradict our preference for privileging youth voice.

Reviewer 3 Report
This study is devoted to one of the topical issues of sustainability, namely the positioning of indigenous peoples in society, and ways of their inclusion in society. However, the title itself indicates two components - politics and poetics. The methodology describes how the phenomenological approach helped the authors of this study. This information is missing in the introduction. The introduction needs to be expanded with a conceptual representation of the political and poetic aspects. Also, the authors need to include such a section as a conclusion. In fact, it was replaced by section 4. Moving Forward. However, it does not negate the need to summarize the results of the study.
Author Response
We wish to thank reviewer 3 for their careful read of the manuscript and for the comments offered.

Reviewer 4 Report
Very powerful research, creative research approach and very well written paper. The privileging of Indigenous youth voice, participation and agency thoughout the research process is outstanding.
Suggestions include:
(1) To further explore and elaborate on the concept of social capital that is referred to on pages 2, 6 and 11. It seems this is readily applicable to the research aim and findings and fits well within an Indigenous epistemology.
(2) The paper could be strengthened by including a section on implications of the research findings for policy and practice, as well as reflections on further research needed.
Author Response
We wish to thank reviewer 4 for their helpful comments.

Round 2
Reviewer 1 Report
The authors did a great work at revising the manuscript. I believe it is ready to accept for publication.